# The Askey–Wilson Integral and Extensions

**Wenchang Chu** 

School of Mathematics and Statistics, Zhoukou Normal University, Zhoukou 466001, China;
hypergeometricx@outlook.com or wenchang.chu@unisalento.it

**Abstract:** By means of the $q$-derivative operator method, we review the $q$-beta integrals of Askey–Wilson and Nassrallah–Rahman. More integrals are evaluated by the author, making use of Bailey's identity of well-poised bilateral $_6\psi_6$-series as well as the extended identity of Karlsson–Minton type for parameterized well-poised bilateral $q$-series.

**Keywords:** basic hypergeometric series; Bailey's identity for well-poised $_6\psi_6$-series; the $q$-derivative operator; Askey–Wilson integral; Nassrallah–Rahman integral

## 1. Introduction and Motivation

For two indeterminates $q$ and $x$, define the shifted factorials by

$$(x;q)_0 = 1 \quad \text{and} \quad (x;q)_n = (1-x)(1-xq)\cdots(1-xq^{n-1}) \quad \text{for} \quad n \in \mathbb{N}.$$

When $|q| < 1$, the following shifted factorial of infinite order is well defined:

$$(x;q)_\infty = \prod_{k=0}^\infty (1-xq^k) \quad \text{and} \quad (x;q)_n = \frac{(x;q)_\infty}{(xq^n;q)_\infty} \quad \text{for} \quad n \in \mathbb{Z}.$$

Its product and quotient forms are abbreviated compactly to

$$
\begin{aligned}
[a, b, \cdots, c; q]_n &= (a;q)_n(b;q)_n\cdots(c;q)_n, \\
\begin{bmatrix} a, b, \cdots, c \\ \alpha, \beta, \cdots, \gamma \end{bmatrix} q \end{bmatrix}_n &= \frac{(a;q)_n(b;q)_n\cdots(c;q)_n}{(\alpha;q)_n(\beta;q)_n\cdots(\gamma;q)_n}.
\end{aligned}
$$

Following Bailey [1] and Gasper–Rahman [2], we define the unilateral and bilateral basic hypergeometric series, respectively, by

$$
{}_{1+r}\phi_s \begin{bmatrix} a_0, a_1, \cdots, a_r \\ b_1, \cdots, b_s \end{bmatrix} q; z \end{bmatrix} = \sum_{n=0}^\infty z^n \begin{bmatrix} a_0, a_1, \cdots, a_r \\ q, b_1, \cdots, b_s \end{bmatrix} q \end{bmatrix}_n,
$$

$$
{}_r\psi_s \begin{bmatrix} a_1, a_2, \cdots, a_r \\ b_1, b_2, \cdots, b_s \end{bmatrix} q; z \end{bmatrix} = \sum_{n=-\infty}^\infty z^n \begin{bmatrix} a_1, a_2, \cdots, a_r \\ b_1, b_2, \cdots, b_s \end{bmatrix} q \end{bmatrix}_n,
$$

where the base $q$ will be confined, throughout the paper, to $0 < |q| < 1$ for nonterminating $q$-series.

In 1985, Askey and Wilson [3] discovered the following remarkable $q$-beta integral formula, (see also [4] (Chapter 10) and [2] (Chapter 6))

$$\int_0^\pi \frac{h(\cos 2\theta; 1)}{h(\cos\theta; a, b, c, d)} \mathrm{d}\theta = \frac{2\pi(abcd;q)_\infty}{[q, ab, ac, ad, bc, bd, cd; q]_\infty}, \tag{1}$$

where the modified Jacobi theta function reads as

$$h(\cos\theta; x) = (xe^{i\theta}, xe^{-i\theta}; q)_\infty = \prod_{n=0}^{\infty} \left(1 - 2q^n x \cos\theta + q^{2n} x^2\right),$$

with its multiparameter form being denoted compactly by

$$h(\cos\theta; \alpha, \beta, \cdots, \gamma) = h(\cos\theta; \alpha) h(\cos\theta; \beta) \cdots h(\cos\theta; \gamma).$$

Due to wide applications to special functions and orthogonal polynomials, this integral formula spurred numerous further works. Refer to [5–8] for different proofs and [9–14] for more extensions, as well as [2] (Chapter 6) and [3,4,13,15] for applications.

The purpose of this article is twofold: reviewing the $q$-beta integrals of Askey–Wilson [3] and Nassrallah–Rahman [10] and establishing several new integral identities, exclusively by employing the $q$-derivative operator approach.

The rest of the paper will be organized as follows. In the next section, the background materials concerning the $q$-derivative operator will be collected. Then, in the third section, the Askey–Wilson integral will be recovered by applying the operator $\mathcal{P}_\lambda(\mathcal{D})$. The new operator $\mathcal{G}_\lambda(\mathcal{E})$ will be introduced in the fourth section, where it is shown how to utilize this operator to boost numerator parameters in $q$-beta integrals. In the fifth section, a novel proof is presented for the integral formula of Nassrallah–Rahman [10] by making use of $\mathcal{P}_\lambda(\mathcal{D})$. A very general new integral formula is derived in the sixth section, where Bailey's identity for well-poised bilateral ${}_6\psi_6$-series plays a crucial role. Most of the new integral identities will finally appear in the seventh section. They are accomplished by combining Bailey's aforementioned fundamental identity with the extended identity of Karlsson–Minton type due to the author [16] for parameterized well-poised bilateral $q$-series.

## 2. Preliminaries about $q$-Derivative Operators

The $q$-derivative operator is a useful tool for proving $q$-series identities (cf. Carlitz [17], Chu [18] and Liu [19]). It is defined by

$$\mathcal{D}_x f(x) := \frac{f(x) - f(qx)}{x} \quad \text{and} \quad \mathcal{D}^n f = \mathcal{D}(\mathcal{D}^{n-1})f \quad \text{for} \quad n = 2, 3, \cdots,$$

with the convention that $\mathcal{D}_x^0 f(x) = f(x)$ for the identity operator. One can show, by means of the induction principle, the following explicit formula

$$\begin{aligned} \mathcal{D}_x^n f(x) &= x^{-n} \sum_{k=0}^{n} q^k \frac{(q^{-n}; q)_k}{(q; q)_k} f(q^k x) \\ &= x^{-n} \sum_{k=0}^{n} (-1)^k \begin{bmatrix} n \\ k \end{bmatrix} q^{\binom{k+1}{2} - nk} f(q^k x) \end{aligned} \tag{2}$$

and the Leibniz rule for the product of two functions

$$\mathcal{D}_x^n \{ f(x) g(x) \} = \sum_{k=0}^{n} q^{k(k-n)} \begin{bmatrix} n \\ k \end{bmatrix} \mathcal{D}_x^k f(x)\, \mathcal{D}_x^{n-k} g(q^k x), \tag{3}$$

where the Gauss $q$-binomial coefficient is expressed in terms of $q$-shifted factorials

$$\begin{bmatrix} n \\ k \end{bmatrix} = \frac{(q; q)_n}{(q; q)_k (q; q)_{n-k}}.$$

There is another "backward" $q$-derivative operator defined by

$$\delta_x f(x) := \frac{f(x) - f(x/q)}{x/q} \quad \text{and} \quad \delta^n f = \delta(\delta^{n-1})f \quad \text{for} \quad n = 2, 3, \cdots,$$

which is related to $\mathcal{D}_x$ by

$$\delta_x f(x) = q \times \left\{ \mathcal{D}_x f(x) \big|_{q \to q^{-1}} \right\}.$$

It is not hard to check that this operator also admits the explicit formula

$$
\begin{aligned}
\delta_x^n f(x) &= (q/x)^n \sum_{k=0}^{n} q^{nk} \frac{(q^{-n}; q)_k}{(q; q)_k} f(q^{-k} x) \\
&= (q/x)^n \sum_{k=0}^{n} (-1)^k \begin{bmatrix} n \\ k \end{bmatrix} q^{\binom{k}{2}} f(q^{-k} x)
\end{aligned}
\tag{4}
$$

and the Leibniz rule for the product of two functions

$$\delta_x^n \{ f(x) g(x) \} = \sum_{k=0}^{n} \begin{bmatrix} n \\ k \end{bmatrix} \delta_x^k f(x) \, \delta_x^{n-k} g(q^{-k} x). \tag{5}$$

When working on generating functions of classical partitions, Rogers [20] introduced, one century ago, the following operators:

$$\mathcal{P}_\lambda(\mathcal{D}) := \sum_{n=0}^{\infty} \frac{\lambda^n}{(q; q)_n} \mathcal{D}^n \quad \text{and} \quad \mathcal{Q}_\lambda(\delta) := \sum_{n=0}^{\infty} \frac{q^{\binom{n}{2}} (-\lambda)^n}{(q; q)_n} \delta^n. \tag{6}$$

They correspond, respectively, to operators $T(\lambda \mathcal{D})$ and $E(\lambda \theta)$ defined in [21,22]. Then it is routine to check that (see Rogers [20]; see also Liu [23])

$$\mathcal{P}_\lambda(\mathcal{D}) \frac{1}{(ax; q)_\infty} = \frac{1}{[ax, a\lambda; q]_\infty}, \tag{7}$$

$$\mathcal{P}_\lambda(\mathcal{D}) \frac{1}{[ax, bx; q]_\infty} = \frac{(\lambda abx; q)_\infty}{[ax, bx, a\lambda, b\lambda; q]_\infty}, \tag{8}$$

$$\mathcal{P}_\lambda(\mathcal{D}) \frac{1}{[ax, bx, cx; q]_\infty} = \begin{bmatrix} \lambda acx \\ ax, bx, cx, a\lambda, c\lambda \end{bmatrix} q \Big]_2 \phi_1 \begin{bmatrix} ax, cx \\ \lambda acx \end{bmatrix} q; b\lambda \Big], \tag{9}$$

where Rogers missed $a\lambda$ in the last denominator.

Recall the bivariate Rogers–Szegö polynomials

$$p_m(b, d, x) = \sum_{k=0}^{m} \begin{bmatrix} m \\ k \end{bmatrix} (bx; q)_k b^{m-k} d^k.$$

They have the following generating function

$$\sum_{m=0}^{\infty} \frac{\lambda^m}{(q; q)_m} p_m(b, d, x) = \begin{bmatrix} \lambda b d x \\ \lambda b, \lambda d \end{bmatrix} q \Big]_\infty$$

and satisfy the recurrence relation

$$x p_{m+1}(b, d, x) = p_m(b, d, x) - (1 - bx)(1 - dx) p_m(b, d, qx).$$

Then by induction on $m$, it is not hard to prove the following two $q$-differentiation formulae (cf. [18,23]):

$$\mathcal{D}^m \frac{1}{[bx, dx; q]_\infty} = \frac{p_m(b, d, x)}{[bx, dx; q]_\infty},$$

$$\mathcal{D}^m \frac{(ex; q)_\infty}{(cx; q)_\infty} = c^m \frac{(q^m ex; q)_\infty}{(cx; q)_\infty} (e/c; q)_m.$$

We remark that these two formulae can also be shown by combining (2) with the second $q$-Chu–Vandermonde formula (cf. [2] (Equation II-6))

$$
{}_2\phi_1\left[\begin{matrix} q^{-m},\, a \\ c \end{matrix}\middle|\, q;\, q\right] = a^m \frac{(c/a;q)_m}{(c;q)_m}.
$$

By making use of the Leibniz rule, we can further compute

$$
\begin{aligned}
\mathcal{P}_\lambda(\mathcal{D})\left[\begin{matrix} xy \\ ax,\, bx,\, cx \end{matrix}\middle|\, q\right]_\infty &= \sum_{n=0}^\infty \frac{\lambda^n}{(q;q)_n} \mathcal{D}^n \left[\begin{matrix} xy \\ ax,\, bx,\, cx \end{matrix}\middle|\, q\right]_\infty \\
&= \sum_{n=0}^\infty \frac{\lambda^n}{(q;q)_n} \sum_{i=0}^n q^{i(i-n)} \begin{bmatrix} n \\ i \end{bmatrix} \mathcal{D}^i \frac{1}{(ax;q)_\infty(bx;q)_\infty} \mathcal{D}^{n-i} \frac{(q^i xy;q)_\infty}{(q^i cx;q)_\infty} \\
&= \sum_{i,j=0}^\infty \frac{\lambda^{i+j} c^j}{(q;q)_i(q;q)_j} \frac{p_i(a,b,x)}{(ax;q)_\infty(bx;q)_\infty} \frac{(q^{i+j}xy;q)_\infty}{(q^i cx;q)_\infty}(y/c;q)_j,
\end{aligned} \tag{10}
$$

where the last line has been justified by the replacement $i + j = n$ on summation indices. Keeping in mind that $|q| < 1$, we can easily check that the following factorial quotient appearing in (10) is bounded, i.e., there exists a $U \in \mathbb{R}$ such that

$$
\left| \frac{1}{(q;q)_i(q;q)_j} \times \frac{(q^{i+j}xy;q)_\infty}{(ax;q)_\infty(bx;q)_\infty(q^i cx;q)_\infty} \right| < U.
$$

Since the $q$-binomial coefficients $\begin{bmatrix} i \\ k \end{bmatrix}$, for all $i, k \in \mathbb{N}_0$, are bounded (say $V$), we can estimate Rogers–Szegö polynomials by

$$
\begin{aligned}
|p_i(a,b,x)| &\le \sum_{k=0}^i \left| \begin{bmatrix} i \\ k \end{bmatrix} (ax;q)_k a^{i-k} b^k \right| \le V \sum_{k=0}^i \eta^k |a|^{i-k}|b|^k \\
&= V \frac{|a|^{i+1} - |b\eta|^{i+1}}{|a| - |b\eta|} \quad \text{where} \quad \eta = \max_{k\in\mathbb{N}_0}\left\{|1 - q^k ax|\right\}.
\end{aligned}
$$

Moreover, we have further

$$
|(y/c;q)_j| = \left| \prod_{k=0}^{j-1}(1 - q^k y/c) \right| \le \rho^j, \quad \text{where} \quad \rho = \max_{k\in\mathbb{N}_0}\left\{|1 - q^k y/c|\right\}.
$$

Summing up, the double series in (10) is dominated by

$$
UV \sum_{i,j=0}^\infty \frac{|a|^{i+1} - |b\eta|^{i+1}}{|a| - |b\eta|} |\lambda|^{i+j} c^j \rho^j.
$$

This is a convergent series as long as $|\lambda|$ is sufficiently small. Therefore, under the same condition, the double series (10) converges absolutely, which allows one to freely exchange the summation order.

Now, by applying the Heine transformation (cf. [2] (Equation III-2))

$$
{}_2\phi_1\left[\begin{matrix} a,\, b \\ c \end{matrix}\middle|\, q;\, z\right] = \left[\begin{matrix} c/a,\, az \\ c,\, z \end{matrix}\middle|\, q\right]_\infty \times {}_2\phi_1\left[\begin{matrix} a,\, abz/c \\ az \end{matrix}\middle|\, q;\, \frac{c}{a}\right], \tag{11}
$$

we can reformulate the above double sum as follows:

$$\begin{bmatrix} xy \\ ax, bx, cx \end{bmatrix} q \Big]_\infty \sum_{i=0}^\infty \lambda^i \frac{p_i(a,b,x)}{(q;q)_i} \frac{(cx;q)_i}{(xy;q)_i} {}_2\phi_1 \begin{bmatrix} y/c, 0 \\ q^i xy \end{bmatrix} q; \lambda c \Big]$$

$$= \begin{bmatrix} \lambda y \\ ax, bx, \lambda c \end{bmatrix} q \Big]_\infty \sum_{i=0}^\infty \lambda^i \frac{p_i(a,b,x)}{(q;q)_i} {}_2\phi_1 \begin{bmatrix} y/c, 0 \\ \lambda y \end{bmatrix} q; q^i cx \Big].$$

Writing the ${}_2\phi_1$ series explicitly and then interchanging the summation order (provided that $|\lambda|$ is sufficiently small), we can reduce the double sum to a single one

$$\mathcal{P}_\lambda(\mathcal{D}) \begin{bmatrix} xy \\ ax, bx, cx \end{bmatrix} q \Big]_\infty = \begin{bmatrix} \lambda y \\ ax, bx, \lambda c \end{bmatrix} q \Big]_\infty \sum_{j=0}^\infty \frac{(y/c;q)_j (cx)^j}{(q;q)_j(\lambda y;q)_j} \sum_i (q^j\lambda)^i \frac{p_i(a,b,x)}{(q;q)_i}$$

$$= \begin{bmatrix} \lambda y \\ ax, bx, \lambda c \end{bmatrix} q \Big]_\infty \sum_{j=0}^\infty \frac{(y/c;q)_j (cx)^j}{(q;q)_j(\lambda y;q)_j} \begin{bmatrix} q^j\lambda abx \\ q^j\lambda a, q^j\lambda b \end{bmatrix} q \Big]_\infty,$$

where we have utilized the generating function of bivariate Rogers–Szegö polynomials. Therefore, we have rigorously proven the following formula

$$\mathcal{P}_\lambda(\mathcal{D}) \begin{bmatrix} xy \\ ax, bx, cx \end{bmatrix} q \Big]_\infty = \begin{bmatrix} \lambda y, \lambda abx \\ ax, bx, \lambda a, \lambda b, \lambda c \end{bmatrix} q \Big]_\infty {}_3\phi_2 \begin{bmatrix} \lambda a, \lambda b, y/c \\ \lambda y, \lambda abx \end{bmatrix} q; cx \Big], \qquad (12)$$

provided that $|\lambda|$ is sufficiently small. This restriction can be removed by analytical continuation.

In view of the Sears transformation (cf. Gasper–Rahman [2] (Equation III-9))

$$_3\phi_2 \begin{bmatrix} a, c, e \\ b, d \end{bmatrix} q; \frac{bd}{ace} \Big] = \begin{bmatrix} b/a, bd/ce \\ b, bd/ace \end{bmatrix} q \Big]_\infty \times {}_3\phi_2 \begin{bmatrix} a, d/c, d/e \\ d, bd/ce \end{bmatrix} q; \frac{b}{a} \Big], \qquad (13)$$

our formula (12) is equivalent to the formula found by Liu [23] (Equation 3.15):

$$\mathcal{P}_\lambda(\mathcal{D}) \frac{(xy;q)_\infty}{[ax, bx, cx;q]_\infty} = \begin{bmatrix} xy, \lambda y, \lambda abcx/y \\ ax, bx, cx, a\lambda, b\lambda, c\lambda \end{bmatrix} q \Big] {}_3\phi_2 \begin{bmatrix} y/a, y/b, y/c \\ xy, \lambda y \end{bmatrix} q; abc\lambda x/y \Big].$$

For this last expression, the original proof of Liu [23], reproduced by Zhang–Wang [24], is lacunose. Subsequently, Liu [25] provided a correct proof via the operator method, but it is less transparent than ours.

Even though Rogers [20] was perhaps the first person to introduce the $q$-differential operators $\mathcal{P}_\lambda(\mathcal{D})$ and $\mathcal{Q}_\lambda(\delta)$ given in (6), he did not reveal their applications. After more than one century since the publication of Rogers' paper, these operators reappeared in the two papers by Chen and Liu [21,22], who worked out several useful properties and reviewed some known $q$-series identities and the Askey–Wilson integral. Due to the simplicity of applying $q$-differential operators to act on known $q$-series identities to obtain new ones, Chen and Liu's papers attracted several followers (for example [7,23–29]) to explore further applications, mainly covering the following topics:

- $q$-Barnes integrals (cf. [22,23,26]),
- $q$-integrals and Sears transformations (cf. [21,22,24–27,29]),
- $q$-series' summation formulae and transformations (cf. [22–25,27–29]) ,
- bilinear generating functions of Rogers–Szegö polynomials (cf. [22,25,26]), and
- the Askey–Wilson integral and the Nassrallah–Rahman integral and extensions (cf. [7,22,24,26,29,30]).

Because this paper is exclusively devoted to the Askey–Wilson integral and extensions, we shall subsequently examine only the ultimate issue in relation to our findings.

### 3. The *q*-Beta Integral of Askey and Wilson

In this section, three *q*-beta integrals, including (1) due to Askey and Wilson [3], will be reviewed by employing the operator $\mathcal{P}_\lambda(\mathcal{D})$ because it admits, according to (8), the following remarkable property:

$$\mathcal{P}_\lambda(\mathcal{D})\frac{1}{h(\cos\theta; x)} = \frac{(\lambda x; q)_\infty}{h(\cos\theta; \lambda, x)}. \tag{14}$$

Our starting point is the following easier *q*-beta integral:

$$\int_0^\pi \frac{h(\cos 2\theta; 1)}{h(\cos\theta; a, c)}\mathrm{d}\theta = \frac{2\pi}{[q, ac; q]_\infty}. \tag{15}$$

By slightly modifying the approach used in [4] (§10.8), we present a simple evaluation by Ramanujan's $_1\psi_1$-series (cf. [2,31] (Equation II-29))

$$_1\psi_1\left[\begin{array}{c} a \\ c \end{array}\middle| q; z\right] = \sum_{k=-\infty}^{+\infty} \frac{(a; q)_k}{(c; q)_k}z^k = \left[\begin{array}{c} q, az, q/az, c/a \\ c, z, c/az, q/a \end{array}\middle| q\right]_\infty, \tag{16}$$

provided that $|c/a| < |z| < 1$ for convergence. Under the replacements $a \to z/a$, $c \to cz$ and $z \to az$, we can reformulate (16) as

$$\left[\begin{array}{c} q, ac, z^2, 1/z^2 \\ az, a/z, cz, c/z \end{array}\middle| q\right]_\infty = \frac{1-z^{-2}}{1-a/z}\sum_{k\geq 1}\frac{(z/a; q)_k}{(cz; q)_k}(az)^k + \frac{1-z^{-2}}{1-a/z}\sum_{k\leq 0}\frac{(z/a; q)_k}{(cz; q)_k}(az)^k$$

$$= (1-z^2)\sum_{k\geq 0}\frac{(qz/a; q)_k}{(cz; q)_{k+1}}(az)^k + (1-z^{-2})\sum_{k\geq 0}\frac{(q/cz; q)_k}{(a/z; q)_{k+1}}(c/z)^k.$$

Observe that the right hand side of the last equation results in a Laurent series with the constant term equal to 2. Letting $z = e^{i\theta}$ and then integrating across the equation with respect to $\theta$ from $-\pi$ to $\pi$, we obtain the following equality

$$(q; q)_\infty(ac; q)_\infty\int_{-\pi}^\pi \frac{h(\cos 2\theta; 1)}{h(\cos\theta; a, c)}\mathrm{d}\theta = 4\pi, \tag{17}$$

which can be restated equivalently as the *q*-beta integral displayed in (15), because the integrand is an even function.

#### 3.1. q-Beta Integrals from 2 to 3 Free Parameters

Specifying the parameter *c* as the variable *x* in (15)

$$\int_0^\pi \frac{h(\cos 2\theta; 1)}{h(\cos\theta; a, x)}\mathrm{d}\theta = \frac{2\pi}{[q, ax; q]_\infty} \tag{18}$$

and then applying $\mathcal{P}_b(\mathcal{D})$ across the last equation, we obtain, in view of (14), the following *q*-beta integral with three free parameters:

$$\int_0^\pi \frac{h(\cos 2\theta; 1)}{h(\cos\theta; a, b, x)}\mathrm{d}\theta = \frac{2\pi}{[q, ab, ax, bx; q]_\infty}. \tag{19}$$

#### 3.2. q-Beta Integrals from 3 to 4 Free Parameters

Now, once again apply $\mathcal{P}_c(\mathcal{D})$ to the last equation. The left hand side becomes

$$(cx; q)_\infty\int_0^\pi \frac{h(\cos 2\theta; 1)}{h(\cos\theta; a, b, c, x)}\mathrm{d}\theta.$$

According to (8), the corresponding right-hand side can be evaluated by

$$\mathcal{P}_c(\mathcal{D}) \frac{2\pi}{[q, ab, ax, bx; q]_\infty} = \frac{2\pi(abcx; q)_\infty}{[q, ab, ac, bc, ax, bx; q]_\infty}.$$

This leads to the following well-known $q$-beta integral formula discovered by Askey and Wilson [3] (Theorem 2.1) (see also [4] (Chapter 10) and [2] (Chapter 6)):

$$\int_0^\pi \frac{h(\cos 2\theta; 1)}{h(\cos \theta; a, b, c, x)} \mathrm{d}\theta = \frac{2\pi(abcx; q)_\infty}{[q, ab, ac, bc, ax, bx, cx; q]_\infty}. \tag{20}$$

*3.3. q-Beta Integrals from 4 to 5 Free Parameters*

Finally, by further applying $\mathcal{P}_d(\mathcal{D})$ to the equality displayed in (20)

$$\int_0^\pi \frac{h(\cos 2\theta; 1)}{h(\cos \theta; a, b, c, d, x)} \mathrm{d}\theta = \frac{2\pi}{[q, ab, ac, bc, dx; q]_\infty} \times \mathcal{P}_d(\mathcal{D}) \begin{bmatrix} abcx \\ ax, bx, cx \end{bmatrix} q \Big]_\infty$$

and then invoking (12), we recover the following extended Askey–Wilson integral (due to Chen and Liu [22]) with five free parameters:

$$\int_0^\pi \frac{h(\cos 2\theta; 1)}{h(\cos \theta; a, b, c, d, x)} \mathrm{d}\theta = {}_3\phi_2 \begin{bmatrix} ab, ac, bc \\ abcd, abcx \end{bmatrix} q; dx \end{bmatrix}$$
$$\times \frac{2\pi[abcx, abcd; q]_\infty}{[q, ab, ac, ad, bc, bd, cd, ax, bx, cx; q]_\infty}. \tag{21}$$

The proof of the Askey–Wilson integral formula reproduced here shows that the approach of $q$-differential operators is efficient. For this reason, there were several papers (see for example [7,22,24,26,29,30]) dedicated to new proofs of Askey–Wilson integral and extensions. However, all these extensions (or complications) are made by inserting an extra ${}_3\phi_2$-series into the integrands (unlike those of simple products in the present paper), cancelling the elegance of the original formula discovered by Askey and Wilson.

Instead of working with $q$-differential operators, there exist other approaches to deal with integrals of the Askey–Wilson type. Ito–Witte [32] made extensions by generalizing the weight functions for the integrals and resolving linear $q$-difference equations. Liu [33] examined double $q$-integrals related to the Askey–Wilson integral through the series rearrangement. Szablowski [34] generalized Askey–Wilson integrals by expanding the Askey–Wilson density into continuous $q$-Hermite polynomials. For details, the interested readers are invited to refer to these papers and related references cited therein.

## 4. The $q$-Gauss Summation Theorem

According to the $q$-Gauss summation theorem (cf. Gasper–Rahman [2] Equation II-8)

$$_2\phi_1 \begin{bmatrix} a, b \\ c \end{bmatrix} q; \frac{c}{ab} \end{bmatrix} = \begin{bmatrix} c/a, c/b \\ c, c/ab \end{bmatrix} q \Big]_\infty, \tag{22}$$

we introduce another operator

$$\mathcal{G}_\lambda(\mathcal{E}) := \sum_{n=0}^\infty \frac{(\lambda^n / x) \mathcal{E}^n}{(q; q)_n (\lambda x; q)_n}, \tag{23}$$

where $\mathcal{E}$ is the $q$-shifted operator defined by

$$\mathcal{E} f(x) := f(qx) \quad \text{for the formal power series } f(x).$$

The above operator $\mathcal{G}_\lambda(\mathcal{E})$ does not seem to have appeared previously, and is substantially different from Rogers' definition in (6) and the Cauchy operator by Chen and Gu [26].

Now, it is routine to check the expression

$$\mathcal{G}_\lambda(\mathcal{E}) \frac{1}{[ax,cx;q]_\infty} = \frac{1}{[ax,cx;q]_\infty} {}_2\phi_1 \left[ \begin{matrix} ax, \; cx \\ \lambda x \end{matrix} \middle| q; \lambda/x \right].$$

In particular, we have the following useful formula

$$\mathcal{G}_\lambda(\mathcal{E}) \frac{1}{h(\cos\theta; x)} = \frac{h(\cos\theta; \lambda)}{h(\cos\theta; x)[\lambda x, \lambda/x; q]_\infty},$$

which will enable us to create numerator parameters inside $q$-beta integrals.

*4.1. The First Integral with Parameter $\lambda$ in Numerator*

By applying $\mathcal{G}_\lambda(\mathcal{E})$ to (18), we have the integral formula:

$$\int_0^\pi \frac{h(\cos 2\theta; 1)h(\cos\theta; \lambda)}{h(\cos\theta; a, x)} d\theta = \frac{2\pi[\lambda x, \lambda/x; q]_\infty}{[q, ax; q]_\infty} \times {}_2\phi_1 \left[ \begin{matrix} 0, \; ax \\ \lambda x \end{matrix} \middle| q; \frac{\lambda}{x} \right].$$

According to (11), we can reformulate this formula by exchanging $a$ and $x$:

$$\int_0^\pi \frac{h(\cos 2\theta; 1)h(\cos\theta; \lambda)}{h(\cos\theta; a, x)} d\theta = \frac{2\pi[\lambda a, \lambda/a; q]_\infty}{[q, ax; q]_\infty} \times {}_2\phi_1 \left[ \begin{matrix} 0, \; ax \\ \lambda a \end{matrix} \middle| q; \frac{\lambda}{a} \right]. \tag{24}$$

*4.2. The Second Integral with Parameter $\lambda$ in Numerator*

If we apply $\mathcal{G}_\lambda(\mathcal{E})$ to (19), we obtain another integral formula

$$\int_0^\pi \frac{h(\cos 2\theta; 1)h(\cos\theta; \lambda)}{h(\cos\theta; a, b, x)} d\theta = \frac{2\pi[\lambda x, \lambda/x; q]_\infty}{[q, ab, ax, bx; q]_\infty} \times {}_2\phi_1 \left[ \begin{matrix} ax, \; bx \\ \lambda x \end{matrix} \middle| q; \frac{\lambda}{x} \right],$$

which can be analogously restated, in view of (11), as

$$\int_0^\pi \frac{h(\cos 2\theta; 1)h(\cos\theta; \lambda)}{h(\cos\theta; a, b, x)} d\theta = \frac{2\pi[\lambda a, \lambda/a; q]_\infty}{[q, ab, ax, bx; q]_\infty} \times {}_2\phi_1 \left[ \begin{matrix} ab, \; ax \\ \lambda a \end{matrix} \middle| q; \frac{\lambda}{a} \right].$$

*4.3. The Third Integral with Parameter $\lambda$ in Numerator*

Finally, applying $\mathcal{G}_\lambda(\mathcal{E})$ to (20) yields the following integral formula:

$$\int_0^\pi \frac{h(\cos 2\theta; 1)h(\cos\theta; \lambda)}{h(\cos\theta; a, b, c, x)} d\theta = \frac{2\pi[abcx, \lambda x, \lambda/x; q]_\infty}{[q, ab, ac, bc, ax, bx, cx; q]_\infty} {}_3\phi_2 \left[ \begin{matrix} ax, \; bx, \; cx \\ abcx, \; \lambda x \end{matrix} \middle| q; \frac{\lambda}{x} \right].$$

By means of (13), the last formula can be rewritten, under the exchange $a \rightleftharpoons x$, as

$$\int_0^\pi \frac{h(\cos 2\theta; 1)h(\cos\theta; \lambda)}{h(\cos\theta; a, b, c, x)} d\theta = \frac{2\pi[abcx, \lambda a, \lambda/a; q]_\infty}{[q, ab, ac, bc, ax, bx, cx; q]_\infty} {}_3\phi_2 \left[ \begin{matrix} ab, \; ac, \; ax \\ \lambda a, \; abcx \end{matrix} \middle| q; \frac{\lambda}{a} \right]. \tag{25}$$

When $\lambda = 0$, both integrals recover the integral (1) of Askey–Wilson.

## 5. The *q*-Beta Integral of Nassrallah and Rahman

In 1985, Nassrallah and Rahman [10] found the following important generalization of the Askey–Wilson integral

$$\int_0^\pi \frac{h(\cos 2\theta; 1)h(\cos\theta; \lambda)}{h(\cos\theta; a, b, c, d, x)}\mathrm{d}\theta = {}_8W_7(\lambda bcd/q : bc, bd, cd, \lambda/a, \lambda/x; ax)$$

$$\times 2\pi \begin{bmatrix} abcd, bcdx, \lambda b, \lambda c, \lambda d \\ q, ab, ac, ad, bc, bd, cd, bx, cx, dx, \lambda bcd \end{bmatrix} q \Big]_\infty, \qquad (26)$$

where, for brevity, the ${}_8W_7$-notation stands for the following well-poised series:

$${}_8W_7(\lambda : a, b, c, d, e; \frac{q\lambda^2}{abcde}) = {}_8\phi_7 \begin{bmatrix} \lambda, \pm q\sqrt{\lambda}, & a, & b, & c, & d, & e \\ \pm\sqrt{\lambda}, & q\lambda/a, q\lambda/b, q\lambda/c, q\lambda/d, q\lambda/e \end{bmatrix} q; \frac{q\lambda^2}{abcde} \Big].$$

This section will be devoted to a new proof of (26). By applying, respectively, $\mathcal{G}_\lambda(\mathcal{E})$ to (21) and $\mathcal{P}_\lambda(\mathcal{D})$ to (25), we shall transform this integral into double sums, which will be reduced, in turn, to the above well-poised ${}_8\phi_7$-series.

### 5.1. The First Double Sum Expression

By applying the operator $\mathcal{G}_\lambda(\mathcal{E})$ to (21), we obtain

$$\int_0^\pi \frac{h(\cos 2\theta; 1)h(\cos\theta; \lambda)}{h(\cos\theta; a, b, c, d, x)}\mathrm{d}\theta = \frac{2\pi[abcd, \lambda x, \lambda/x; q]_\infty}{[q, ab, ac, ad, bc, bd, cd; q]_\infty}$$

$$\times \mathcal{G}_\lambda(\mathcal{E}) \begin{bmatrix} abcx \\ ax, bx, cx \end{bmatrix} q \Big]_\infty {}_3\phi_2 \begin{bmatrix} ab, ac, bc \\ abcd, abcx \end{bmatrix} q; dx \Big].$$

Observing that

$$\mathcal{G}_\lambda(\mathcal{E}) \begin{bmatrix} abcx \\ ax, bx, cx \end{bmatrix} q \Big]_\infty {}_3\phi_2 \begin{bmatrix} ab, ac, bc \\ abcd, abcx \end{bmatrix} q; dx \Big]$$

$$= \sum_{i,j\geq 0} \frac{(\lambda/x)^i d^j}{[q, \lambda x; q]_i} \begin{bmatrix} ab, ac, bc \\ q, abcd \end{bmatrix} q \Big]_j \mathcal{E}^i \begin{bmatrix} q^j abcx \\ ax, bx, cx \end{bmatrix} q \Big]_\infty x^j$$

$$= \begin{bmatrix} abcx \\ ax, bx, cx \end{bmatrix} q \Big]_\infty \sum_{i,j\geq 0} \frac{q^{ij}(\lambda/x)^i(dx)^j}{(abcx; q)_{i+j}} \begin{bmatrix} ax, bx, cx \\ q, \lambda x \end{bmatrix} q \Big]_i \begin{bmatrix} ab, ac, bc \\ q, abcd \end{bmatrix} q \Big]_j,$$

we derive the first double sum expression

$$\int_0^\pi \frac{h(\cos 2\theta; 1)h(\cos\theta; \lambda)}{h(\cos\theta; a, b, c, d, x)}\mathrm{d}\theta = 2\pi \begin{bmatrix} abcd, abcx, \lambda x, \lambda/x \\ q, ab, ac, ad, bc, bd, cd, ax, bx, cx \end{bmatrix} q \Big]_\infty$$

$$\times \sum_{i,j\geq 0} \frac{q^{ij}}{(abcx; q)_{i+j}} \begin{bmatrix} ax, bx, cx \\ q, \lambda x \end{bmatrix} q \Big]_i \left(\frac{\lambda}{x}\right)^i \begin{bmatrix} ab, ac, bc \\ q, abcd \end{bmatrix} q \Big]_j (dx)^j. \qquad (27)$$

### 5.2. The Second Double Sum Expression

Alternatively, by applying $\mathcal{P}_d(\mathcal{D})$ to (25), we have

$$\int_0^\pi \frac{h(\cos 2\theta; 1)h(\cos\theta; \lambda)}{h(\cos\theta; a, b, c, d, x)}\mathrm{d}\theta = \frac{2\pi[\lambda a, \lambda/a; q]_\infty}{[q, ab, ac, bc, dx; q]_\infty}$$

$$\times \mathcal{P}_d(\mathcal{D}) \begin{bmatrix} abcx \\ ax, bx, cx \end{bmatrix} q \Big]_\infty {}_3\phi_2 \begin{bmatrix} ab, ac, ax \\ \lambda a, abcx \end{bmatrix} q; \frac{\lambda}{a} \Big].$$

By means of (12), we can evaluate

$$
\mathcal{P}_d(\mathcal{D})\begin{bmatrix} abcx \\ ax, bx, cx \end{bmatrix}q\Big]_\infty {}_3\phi_2\begin{bmatrix} ab,\, ac,\, ax \\ \lambda a,\, abcx \end{bmatrix}q; \frac{\lambda}{a}\Big]
$$

$$
= \sum_{i\ge 0}\begin{bmatrix} ab, ac \\ q, \lambda a \end{bmatrix}q\Big]_i \Big(\frac{\lambda}{a}\Big)^i \mathcal{P}_d(\mathcal{D})\begin{bmatrix} q^i abcx \\ q^i ax, bx, cx \end{bmatrix}q\Big]_\infty
$$

$$
= \sum_{i\ge 0}\begin{bmatrix} ab, ac \\ q, \lambda a \end{bmatrix}q\Big]_i \Big(\frac{\lambda}{a}\Big)^i \begin{bmatrix} q^i abcd, q^i abdx \\ q^i ax, bx, q^i ad, bd, cd \end{bmatrix}q\Big]_\infty {}_3\phi_2\begin{bmatrix} q^i ab,\, q^i ad,\, bd \\ q^i abcd,\, q^i abdx \end{bmatrix}q; cx\Big]
$$

$$
= \begin{bmatrix} abcd, abdx \\ ad, bd, cd, ax, bx \end{bmatrix}q\Big]_\infty \sum_{i,j\ge 0}\begin{bmatrix} ab, ad \\ abcd, abdx \end{bmatrix}q\Big]_{i+j}\begin{bmatrix} ac, ax \\ q, \lambda a \end{bmatrix}q\Big]_i \Big(\frac{\lambda}{a}\Big)^i \frac{(bd; q)_j}{(q; q)_j}(cx)^j,
$$

which leads to another double sum expression

$$
\int_0^\pi \frac{h(\cos 2\theta; 1) h(\cos\theta; \lambda)}{h(\cos\theta; a, b, c, d, x)}\,\mathrm{d}\theta = 2\pi\begin{bmatrix} abcd, abdx, \lambda a, \lambda/a \\ q, ab, ac, bc, ad, bd, cd, ax, bx, dx \end{bmatrix}q\Big]_\infty
$$

$$
\times \sum_{i,j\ge 0}\begin{bmatrix} ab, ad \\ abcd, abdx \end{bmatrix}q\Big]_{i+j}\begin{bmatrix} ac, ax \\ q, \lambda a \end{bmatrix}q\Big]_i \Big(\frac{\lambda}{a}\Big)^i \frac{(bd; q)_j}{(q; q)_j}(cx)^j. \tag{28}
$$

### 5.3. Reduction of Double Sum to Single One

Next, we are going to simplify the last double sum. Recalling the nonterminating well-poised $_6\phi_5$-series (cf. Gasper–Rahman [2] (Equation II-20))

$$
{}_6\phi_5\begin{bmatrix} a,\, q\sqrt{a},\, -q\sqrt{a},\, b,\, c,\, d \\ \sqrt{a},\, -\sqrt{a},\, qa/b,\, qa/c,\, qa/d \end{bmatrix}q; \frac{qa}{bcd}\Big] = \begin{bmatrix} qa, qa/bc, qa/bd, qa/cd \\ qa/b, qa/c, qa/d, qa/bcd \end{bmatrix}q\Big]_\infty,
$$

we have, as a limiting case, the following equality:

$$
\begin{bmatrix} cx, q^{2i+2j}a^2 bcdx \\ q^{i+j}abcx, q^{i+j}acdx \end{bmatrix}q\Big]_\infty = \sum_{k\ge 0}\frac{1 - q^{2i+2j+2k-1}a^2 bcdx}{1 - q^{2i+2j-1}a^2 bcdx} q^{\binom{k}{2}}
$$

$$
\times \begin{bmatrix} q^{2i+2j-1}a^2 bcdx, q^{i+j}ab, q^{i+j}ad \\ q, q^{i+j}abcx, q^{i+j}acdx \end{bmatrix}q\Big]_k (-cx)^k.
$$

By inserting this relation inside the double double sum in (28), we can reformulate the triple sum, by letting $k + j = n$, as follows:

$$
\Omega = \sum_{i,j\ge 0}\begin{bmatrix} ab, ad \\ abcd, abdx \end{bmatrix}q\Big]_{i+j}\begin{bmatrix} ac, ax \\ q, \lambda a \end{bmatrix}q\Big]_i \Big(\frac{\lambda}{a}\Big)^i \frac{(bd; q)_j}{(q; q)_j}(cx)^j
$$

$$
\times \begin{bmatrix} q^{i+j}abcx, q^{i+j}acdx \\ cx, q^{2i+2j}a^2 bcdx \end{bmatrix}q\Big]_\infty \sum_{k\ge 0}\frac{1 - q^{2i+2j+2k-1}a^2 bcdx}{1 - q^{2i+2j-1}a^2 bcdx}
$$

$$
\times q^{\binom{k}{2}}\begin{bmatrix} q^{2i+2j-1}a^2 bcdx, q^{i+j}ab, q^{i+j}ad \\ q, q^{i+j}abcx, q^{i+j}acdx \end{bmatrix}q\Big]_k (-cx)^k
$$

$$
= \begin{bmatrix} abcx, acdx \\ cx, a^2 bcdx \end{bmatrix}q\Big]_\infty \sum_{i,j,k\ge 0}\begin{bmatrix} ab, ad \\ abcx, acdx \end{bmatrix}q\Big]_{i+j+k}\begin{bmatrix} ac, ax \\ q, \lambda a \end{bmatrix}q\Big]_i \Big(\frac{\lambda}{a}\Big)^i
$$

$$
\times (-1)^k \frac{(a^2 bcdx/q; q)_{2i+2j+k}}{[abcd, abdx; q]_{i+j}}\frac{1 - q^{2i+2j+2k-1}a^2 bcdx}{1 - q^{-1}a^2 bcdx}\frac{(cx)^{j+k}(bd; q)_j}{(q; q)_j (q; q)_k} q^{\binom{k}{2}}
$$

$$
= \begin{bmatrix} abcx, acdx \\ cx, a^2 bcdx \end{bmatrix}q\Big]_\infty \sum_{i,n\ge 0}\frac{(-cx)^n}{(q; q)_n} q^{\binom{n}{2}}\begin{bmatrix} ab, ad \\ abcx, acdx \end{bmatrix}q\Big]_{i+n}\begin{bmatrix} ac, ax \\ q, \lambda a \end{bmatrix}q\Big]_i \Big(\frac{\lambda}{a}\Big)^i
$$

$$
\times \frac{1 - q^{2i+2n-1}a^2 bcdx}{1 - q^{-1}a^2 bcdx}\frac{(a^2 bcdx/q; q)_{2i+n}}{[abcd, abdx; q]_i} {}_3\phi_2\begin{bmatrix} q^{-n}, bd, q^{2i+n-1}a^2 bcdx \\ q^i abcd, q^i abdx \end{bmatrix}q; q\Big].
$$

Evaluating the last $_3\phi_2$-series by the $q$-Pfaff–Saalschˊutz theorem (cf. [2] (Equation II-12))

$$_3\phi_2\left[\begin{array}{c}q^{-n},bd,q^{2i+n-1}a^2bcdx\\q^iabcd,q^iabdx\end{array}\bigg|q;q\right]=(bd)^n\left[\begin{array}{c}q^iac,q^iax\\q^iabcd,q^iabdx\end{array}\bigg|q\right]_n,$$

we obtain the following transformation of the double $\Omega$-sum to another one

$$\Omega=\left[\begin{array}{c}abcx,acdx\\cx,a^2bcdx\end{array}\bigg|q\right]_\infty\sum_{i,n\geq0}\frac{1-q^{2i+2n-1}a^2bcdx}{1-q^{-1}a^2bcdx}\left[\begin{array}{c}ab,ac,ad,ax\\abcd,abcx,abdx,acdx\end{array}\bigg|q\right]_{i+n}$$

$$\times q^{\binom{n}{2}}\frac{(-bcdx)^n}{(q;q)_n}\frac{(a^2bcdx/q;q)_{2i+n}}{[q,\lambda a;q]_i}\left(\frac{\lambda}{a}\right)^i$$

$$=\left[\begin{array}{c}abcx,acdx\\cx,a^2bcdx\end{array}\bigg|q\right]_\infty\sum_{m\geq0}\frac{1-q^{2m-1}a^2bcdx}{1-q^{-1}a^2bcdx}\left[\begin{array}{c}ab,ac,ad,ax\\abcd,abcx,abdx,acdx\end{array}\bigg|q\right]_m$$

$$\times\frac{(a^2bcdx/q;q)_m}{(q;q)_m}{}_2\phi_1\left[\begin{array}{c}q^{-m},q^{m-1}a^2bcdx\\\lambda a\end{array}\bigg|q;\frac{q\lambda}{abcdx}\right]q^{\binom{m}{2}}(-bcdx)^m,$$

where the last passage is justified by $i+n=m$. By making use of (22)

$$_2\phi_1\left[\begin{array}{c}q^{-m},q^{m-1}a^2bcdx\\\lambda a\end{array}\bigg|q;\frac{q\lambda}{abcdx}\right]=\frac{(abcdx/\lambda;q)_m}{(\lambda a;q)_m}\left(\frac{-\lambda}{abcdx}\right)^mq^{-\binom{m}{2}},$$

and then simplifying the result, we can finally express the $\Omega$-sum as the following well-poised series:

$$\sum_{i,j\geq0}\left[\begin{array}{c}ab,ad\\abcd,abdx\end{array}\bigg|q\right]_{i+j}\left[\begin{array}{c}ac,ax\\q,\lambda a\end{array}\bigg|q\right]_i\left(\frac{\lambda}{a}\right)^i\frac{(bd;q)_j}{(q;q)_j}(cx)^j$$

$$=\left[\begin{array}{c}abcx,acdx\\cx,a^2bcdx\end{array}\bigg|q\right]_\infty{}_8W_7(a^2bcdx/q:ab,ac,ad,ax,abcdx/\lambda;\lambda/a).\tag{29}$$

By comparing the two expressions displayed as (27) and (28), we obtain a transformation formula for another double sum:

$$\sum_{i,j\geq0}\left[\begin{array}{c}ax,bx,cx\\q,\lambda x\end{array}\bigg|q\right]_i\frac{q^{ij}(\lambda/x)^i(dx)^j}{(abcx;q)_{i+j}}\left[\begin{array}{c}ab,ac,bc\\q,abcd\end{array}\bigg|q\right]_j=\Omega\left[\begin{array}{c}abdx,cx,\lambda a,\lambda/a\\abcx,dx,\lambda x,\lambda/x\end{array}\bigg|q\right]_\infty$$

$$=\left[\begin{array}{c}\lambda a,\lambda/a,abdx,acdx\\\lambda x,\lambda/x,dx,a^2bcdx\end{array}\bigg|q\right]_\infty{}_8W_7(a^2bcdx/q:ab,ac,ad,ax,abcdx/\lambda;\lambda/a).$$

It should be pointed out that (29) is equivalent to the following formula

$$\sum_{i,j=0}^\infty\left[\begin{array}{c}a,c\\b,d\end{array}\bigg|q\right]_{i+j}\left[\begin{array}{c}u\\q\end{array}\bigg|q\right]_i\left[\begin{array}{c}b/u,d/u\\q,\quad v\end{array}\bigg|q\right]_j\left(\frac{bd}{acu}\right)^i\left(\frac{uv}{ac}\right)^j$$

$$=\left[\begin{array}{c}bd/au,bd/cu\\bd/u,bd/acu\end{array}\bigg|q\right]_\infty{}_8\phi_7\left[\begin{array}{c}bd/qu,\pm\sqrt{bdq/u},a,c,b/u,\quad d/u,\quad bd/uv\\\pm\sqrt{bd/qu},b,d,\quad v,\quad bd/au,bd/cu\end{array}\bigg|q;\frac{uv}{ac}\right]\tag{30}$$

by Chu–Zhang [35], who established more summation and reduction formulae for the $q$-Kampé de Fériet Function.

By combining (28) with (29) and then exchanging $a$ and $x$, we prove the following important $q$-beta integral formula (cf. [2] (Equation 6.3.7)):

$$\int_0^\pi \frac{h(\cos 2\theta; 1)h(\cos \theta; \lambda)}{h(\cos \theta; a, b, c, d, x)}\,\mathrm{d}\theta = 2\pi\Omega \left[ \begin{matrix} bcdx, abdx, \lambda x, \lambda/x \\ q, ab, ad, bc, bd, cd, ax, bx, cx, dx \end{matrix} \Big| q \right]_\infty$$

$$= 2\pi \left[ \begin{matrix} abcx, abdx, acdx, bcdx, \lambda x, \lambda/x \\ q, ab, ac, ad, bc, bd, cd, ax, bx, cx, dx, abcdx^2 \end{matrix} \Big| q \right]_\infty \tag{31}$$

$$\times {}_8W_7(abcdx^2/q : ax, bx, cx, dx, abcdx/\lambda; \lambda/x).$$

The special case $\lambda = abcdx$ was rederived in [22], where the $q$-differential operator $\mathcal{P}_\lambda(\mathcal{D})$ was employed. In addition, it is not hard to check that (31) is equivalent to the original formula (26) of Nassrallah and Rahman [10] in view of the following limiting form of Bailey's transformation for well-poised ${}_{10}\phi_9$-series (cf. [2] (Equation 2.10.1))

$$\left[ \begin{matrix} qa, qa/uv, q\lambda/u, q\lambda/v \\ qa/u, qa/v, q\lambda, q\lambda/uv \end{matrix} \Big| q \right]_\infty = \frac{{}_8W_7(a : b, c, d, u, v; q^2a^2/bcduv)}{{}_8W_7(\lambda : \lambda b/a, \lambda c/a, \lambda d/a, u, v; qa/uv)},$$

provided that $\lambda = qa^2/bcd$, $|qa/uv| < 1$ and $|q\lambda/uv| < 1$.

By applying this transformation again to the ${}_8W_7$-series in (26), we recover another expression (cf. [2] (Equation 6.3.9)):

$$\int_0^\pi \frac{h(\cos 2\theta; 1)h(\cos \theta; \lambda)}{h(\cos \theta; a, b, c, d, x)}\,\mathrm{d}\theta = {}_8W_7(\lambda^2/q : \lambda/a, \lambda/b, \lambda/c, \lambda/d, \lambda/x; abcdx/\lambda)$$

$$\times 2\pi \left[ \begin{matrix} \lambda a, \lambda b, \lambda c, \lambda d, \lambda x, abcdx/\lambda \\ q, ab, ac, ad, bc, bd, cd, ax, bx, cx, dx, \lambda^2 \end{matrix} \Big| q \right]_\infty. \tag{32}$$

When $\lambda = x$, both (31) and (32) reduce to Askey–Wilson's formula (1).

## 6. Bailey's Well-Poised Bilateral ${}_6\psi_6$-Series

For the five complex parameters $a, b, c, d$ and $\lambda$ subject to $|q\lambda^2/bcde| < 1$, Bailey [36] (see also [2,6] (Equation II-33)) found the following beautiful formula:

$$_6\psi_6 \left[ \begin{matrix} q\sqrt{\lambda}, -q\sqrt{\lambda}, & a, & b, & c, & d \\ \sqrt{\lambda}, & -\sqrt{\lambda}, & q\lambda/a, & q\lambda/b, & q\lambda/c, & q\lambda/d \end{matrix} \Big| q; \frac{q\lambda^2}{abcd} \right]$$

$$= \frac{(q, q\lambda, q/\lambda, q\lambda/ab, q\lambda/ac, q\lambda/ad, q\lambda/bc, q\lambda/bd, q\lambda/cd; q)_\infty}{(q\lambda/a, q\lambda/b, q\lambda/c, q\lambda/d, q/a, q/b, q/c, q/d, q\lambda^2/abcd; q)_\infty}.$$

Now we define the following bilateral operator

$$\sum_{n=-\infty}^{+\infty} \frac{1 - q^{2n-1}abcdx^2}{1 - abcdx^2/q} \left[ \begin{matrix} abcdx/u, abcdx/v \\ ux, \quad vx \end{matrix} \Big| q \right]_n \left( \frac{uv}{q} \right)^n \mathcal{E}^n$$

and then apply it to the integral

$$\int_0^\pi \frac{h(\cos 2\theta; 1)h(\cos \theta; abcdx)}{h(\cos \theta; a, b, c, d, x)}\,\mathrm{d}\theta$$

$$= 2\pi \left[ \begin{matrix} abcd, abcx, abdx, acdx, bcdx \\ q, ab, ac, ad, bc, bd, cd, ax, bx, cx, dx \end{matrix} \Big| q \right]_\infty, \tag{33}$$

which is, in fact, the case $\lambda = abcdx$ of (31), as noted by Rahman [11].

The left-hand side can be expressed compactly as

$$\left[ \begin{matrix} q, abcd, uv/abcd, abcdx^2, q^2/abcdx^2 \\ uv/q, ux, vx, qu/abcdx, qv/abcdx \end{matrix} \Big| q \right]_\infty \int_0^\pi \frac{h(\cos 2\theta; 1)h(\cos \theta; u, v)}{h(\cos \theta; a, b, c, d, x, q/x)}\,\mathrm{d}\theta.$$

The corresponding right member becomes

$$2\pi \left[ \begin{matrix} abcd, abcx, abdx, acdx, bcdx \\ q, ab, ac, ad, bc, bd, cd, ax, bx, cx, dx \end{matrix} \bigg| q \right]_\infty$$

$$\times {}_8\psi_8 \left[ \begin{matrix} \pm x\sqrt{abcdq}, & ax, & bx, & cx, & dx, & abcdx/u, & abcdx/v \\ \pm x\sqrt{abcd/q}, & bcdx, & acdx, & abdx, & abcx, & ux, & vx \end{matrix} \bigg| q; \frac{uv}{q} \right].$$

Therefore, we have established the following new $q$-beta integral formula

$$\int_0^\pi \frac{h(\cos 2\theta; 1)h(\cos\theta; u, v)}{h(\cos\theta; a, b, c, d, x, q/x)} \, d\theta$$

$$= {}_8\psi_8 \left[ \begin{matrix} \pm x\sqrt{abcdq}, & ax, & bx, & cx, & dx, & abcdx/u, & abcdx/v \\ \pm x\sqrt{abcd/q}, & bcdx, & acdx, & abdx, & abcx, & ux, & vx \end{matrix} \bigg| q; \frac{uv}{q} \right] \tag{34}$$

$$\times 2\pi \left[ \begin{matrix} abcx, abdx, acdx, bcdx, uv/q, ux, vx, qu/abcdx, qv/abcdx \\ q, q, ab, ac, ad, bc, bd, cd, ax, bx, cx, dx, uv/abcd, abcdx^2, q^2/abcdx^2 \end{matrix} \bigg| q \right]_\infty,$$

which contains (31) as the particular case $v = q/x$. This formula can also be considered as a counterpart of the one derived by Rahman [37] (cf. [2] (Page 173)),

$$\int_0^\pi \frac{h(\cos 2\theta; 1)h(\cos\theta; \lambda, q/\lambda)}{h(\cos\theta; c_1, c_2, c_3, c_4, c_5, c_6)} \, d\theta$$

$$= 2\pi \left[ \begin{matrix} c_1 c_2 c_3 c_4 c_5 c_6/q, \{\lambda c_k, qc_k/\lambda\}_{1\le k\le 6} \\ q, q, \lambda^2, q^2/\lambda^2, \{c_i c_j\}_{1\le i<j\le 6} \end{matrix} \bigg| q \right]_\infty \tag{35}$$

$$\times {}_8\psi_8 \left[ \begin{matrix} \pm\lambda\sqrt{q}, & \lambda/c_1, & \lambda/c_2, & \lambda/c_3, & \lambda/c_4, & \lambda/c_5, & \lambda/c_6 \\ \pm\lambda/\sqrt{q}, & \lambda c_1, & \lambda c_2, & \lambda c_3, & \lambda c_4, & \lambda c_5, & \lambda c_6 \end{matrix} \bigg| q; \frac{c_1 c_2 c_3 c_4 c_5 c_6}{q} \right],$$

where a pair of "reciprocal parameters" in the denominator is exchanged with the parameter pair in the numerator.

## 7. The $q$-Beta Integrals of Karlsson–Minton Type

Throughout this section, we shall fix the $n$-pairs of complex parameters $\{u_\iota, v_\iota\}$ subject to the finite conditions

$$u_\iota/v_\iota = q^{m_\iota} \quad \text{with} \quad m_\iota \in \mathbb{N}_0 \quad \text{and} \quad m = \sum_{\iota=1}^n m_\iota$$

as well as

$$U := \prod_{k=1}^n u_k, \quad V := \prod_{k=1}^n v_k \quad \text{and} \quad q^m = U/V.$$

By utilizing a generalized identity (see Chu [16] (Theorem 2) and Chu–Wang [38] (Corollary 6)) of Karlsson–Minton type for the well-poised bilateral series, Chu and Ma [39] found the following general integral formula

$$\int_0^\pi \frac{h(\cos 2\theta; 1)}{h(\cos\theta; a, q/a, b, d)} \prod_{k=1}^n \frac{h(\cos\theta; u_k)}{h(\cos\theta; v_k)} \, d\theta$$

$$= \frac{2\pi/(1 - bdV/U)}{[q, q, ab, ad, qb/a, qd/a; q]_\infty} \prod_{\iota=1}^n \left[ \begin{matrix} au_\iota, qu_\iota/a \\ av_\iota, qv_\iota/a \end{matrix} \bigg| q \right]_\infty. \tag{36}$$

We shall employ it to evaluate further $q$-beta integrals. It should be pointed out that most of the formulae appearing in this section are new.

### 7.1. Boosting the First Numerator Parameter by $\mathcal{G}_\lambda(\mathcal{E})$

Specify the parameter $d$ as the variable $x$ in the last equation. When applying $\mathcal{P}_\lambda(\mathcal{D})$ to the resulting equation, there is no simple expression for the right member. However, if we apply $\mathcal{G}_\lambda(\mathcal{E})$ to the same equation, we obtain the following one:

$$
\begin{aligned}
&\int_0^\pi \frac{h(\cos 2\theta;1)h(\cos\theta;\lambda)}{h(\cos\theta;a,q/a,b,x)}\prod_{k=1}^n \frac{h(\cos\theta;u_k)}{h(\cos\theta;v_k)}d\theta\\
&={}_3\phi_2\left[\begin{matrix} ax,qx/a,q^{-m}bx \\ \lambda x,qbxV/U \end{matrix}\bigg|q;\frac{\lambda}{x}\right]\prod_{l=1}^n\left[\begin{matrix} au_l,qu_l/a \\ av_l,qv_l/a \end{matrix}\bigg|q\right]_\infty\\
&\quad\times\frac{2\pi}{1-bxV/U}\left[\begin{matrix} \lambda x,\quad \lambda/x \\ q,q,ab,qb/a,ax,qx/a \end{matrix}\bigg|q\right]_\infty.
\end{aligned}
\tag{37}
$$

By making use of (13), we can restate (37) as another new formula

$$
\begin{aligned}
&\int_0^\pi \frac{h(\cos 2\theta;1)h(\cos\theta;\lambda)}{h(\cos\theta;a,q/a,b,x)}\prod_{k=1}^n \frac{h(\cos\theta;u_k)}{h(\cos\theta;v_k)}d\theta\\
&={}_3\phi_2\left[\begin{matrix} \lambda/a,\,\lambda a/q,\,bxV/U \\ \lambda x,\quad \lambda bV/U \end{matrix}\bigg|q;q\right]\prod_{l=1}^n\left[\begin{matrix} au_l,qu_l/a \\ av_l,qv_l/a \end{matrix}\bigg|q\right]_\infty\\
&\quad\times 2\pi\left[\begin{matrix} \lambda x,\quad \lambda bV/U \\ q,ab,qb/a,ax,qx/a,bxV/U \end{matrix}\bigg|q\right]_\infty.
\end{aligned}
\tag{38}
$$

When $\lambda=bV/U$ in (37), we have, in view of the Gauss summation theorem, the following closed expression:

$$
\begin{aligned}
&\int_0^\pi \frac{h(\cos 2\theta;1)h(\cos\theta;bV/U)}{h(\cos\theta;a,q/a,b,x)}\prod_{k=1}^n \frac{h(\cos\theta;u_k)}{h(\cos\theta;v_k)}d\theta\\
&=2\pi\left[\begin{matrix} abV/U,qbV/aU \\ q,q,ab,qb/a,ax,qx/a \end{matrix}\bigg|q\right]_\infty\prod_{l=1}^n\left[\begin{matrix} au_l,qu_l/a \\ av_l,qv_l/a \end{matrix}\bigg|q\right]_\infty.
\end{aligned}
\tag{39}
$$

We remark that a similar formula was obtained in [40] (Theorem 9), where the fraction before the product in the integrand contains only one $h$-function in the numerator and four $h$-functions with free parameters in the denominator. However, their expression on the right-hand side involves very complicated multiple sums.

### 7.2. Boosting Denominator Parameter d by $\mathcal{P}_d(\mathcal{D})$

Now, by applying $\mathcal{P}_d(\mathcal{D})$ to (39), we derive the following integral

$$
\begin{aligned}
&\int_0^\pi \frac{h(\cos 2\theta;1)h(\cos\theta;bV/U)}{h(\cos\theta;a,q/a,b,d,x)}\prod_{k=1}^n \frac{h(\cos\theta;u_k)}{h(\cos\theta;v_k)}d\theta\\
&=\frac{2\pi}{1-dx}\left[\begin{matrix} abV/U,qbV/aU \\ q,q,ab,qb/a,ad,qd/a,ax,qx/a \end{matrix}\bigg|q\right]_\infty\prod_{l=1}^n\left[\begin{matrix} au_l,qu_l/a \\ av_l,qv_l/a \end{matrix}\bigg|q\right]_\infty.
\end{aligned}
\tag{40}
$$

When $b=0$, this reduces to (36), as derived by Chu and Ma [39].

### 7.3. Boosting the Second Numerator Parameter $\lambda$ by $\mathcal{G}_\lambda(\mathcal{E})$

By applying $\mathcal{G}_\lambda(\mathcal{E})$ further to (40), we obtain the integral formula below:

$$
\int_0^\pi \frac{h(\cos 2\theta; 1)h(\cos \theta; \lambda, bV/U)}{h(\cos \theta; a, q/a, b, d, x)} \prod_{k=1}^n \frac{h(\cos \theta; u_k)}{h(\cos \theta; v_k)} d\theta
$$
$$
= \frac{2\pi}{1-dx} \left[ \begin{matrix} abV/U, qbV/aU, \lambda x, \lambda/x \\ q, q, ab, qb/a, ad, qd/a, ax, qx/a \end{matrix} \middle| q \right]_\infty \tag{41}
$$
$$
\times {}_3\phi_2 \left[ \begin{matrix} ax, qx/a, \ dx \\ \lambda x, \ qdx \end{matrix} \middle| q; \frac{\lambda}{x} \right] \prod_{\iota=1}^n \left[ \begin{matrix} au_\iota, qu_\iota/a \\ av_\iota, qv_\iota/a \end{matrix} \middle| q \right]_\infty .
$$

By means of (13), this formula can be restated as

$$
\int_0^\pi \frac{h(\cos 2\theta; 1)h(\cos \theta; \lambda, bV/U)}{h(\cos \theta; a, q/a, b, d, x)} \prod_{k=1}^n \frac{h(\cos \theta; u_k)}{h(\cos \theta; v_k)} d\theta
$$
$$
= 2\pi \left[ \begin{matrix} abV/U, qbV/aU, \lambda d, \lambda x \\ q, ab, qb/a, ad, qd/a, ax, qx/a, dx \end{matrix} \middle| q \right]_\infty \tag{42}
$$
$$
\times {}_3\phi_2 \left[ \begin{matrix} \lambda/a, \lambda a/q, \ dx \\ \lambda d, \ \lambda x \end{matrix} \middle| q; q \right] \prod_{\iota=1}^n \left[ \begin{matrix} au_\iota, qu_\iota/a \\ av_\iota, qv_\iota/a \end{matrix} \middle| q \right]_\infty .
$$

### 7.4. Boosting Another Denominator Parameter $c$ by $\mathcal{P}_c(\mathcal{D})$

Now, by applying $\mathcal{P}_c(\mathcal{D})$ to (42), we have

$$
\int_0^\pi \frac{h(\cos 2\theta; 1)h(\cos \theta; \lambda, bV/U)}{h(\cos \theta; a, q/a, b, c, d, x)} \prod_{k=1}^n \frac{h(\cos \theta; u_k)}{h(\cos \theta; v_k)} d\theta
$$
$$
= 2\pi \left[ \begin{matrix} abV/U, qbV/aU, \lambda d \\ q, ab, qb/a, ad, qd/a, cx \end{matrix} \middle| q \right]_\infty \prod_{\iota=1}^n \left[ \begin{matrix} au_\iota, qu_\iota/a \\ av_\iota, qv_\iota/a \end{matrix} \middle| q \right]_\infty
$$
$$
\times \sum_{i\geq 0} q^i \left[ \begin{matrix} \lambda/a, \lambda a/q \\ q, \lambda d \end{matrix} \middle| q \right]_i \mathcal{P}_c(\mathcal{D}) \left[ \begin{matrix} q^i \lambda x \\ ax, qx/a, q^i dx \end{matrix} \middle| q \right]_\infty .
$$

According to (12), we also have

$$
\mathcal{P}_c(\mathcal{D}) \left[ \begin{matrix} q^i \lambda x \\ ax, qx/a, q^i dx \end{matrix} \middle| q \right]_\infty = \left[ \begin{matrix} q^i \lambda c, q^i \lambda x, qcdx/\lambda \\ ac, qc/a, ax, qx/a, q^i cd, q^i dx \end{matrix} \middle| q \right]_\infty
$$
$$
\times {}_3\phi_2 \left[ \begin{matrix} \lambda/d, q^i \lambda/a, q^{i-1}\lambda a \\ q^i \lambda c, \ q^i \lambda x \end{matrix} \middle| q; \frac{qcdx}{\lambda} \right].
$$

Therefore, we have the following double sum expression:

$$
\sum_{i\geq 0} q^i \left[ \begin{matrix} \lambda/a, \lambda a/q \\ q, \lambda d \end{matrix} \middle| q \right]_i \mathcal{P}_c(\mathcal{D}) \left[ \begin{matrix} \lambda x \\ ax, qx/a, dx \end{matrix} \middle| q \right]_\infty
$$
$$
= \left[ \begin{matrix} \lambda c, \lambda x, qcdx/\lambda \\ ac, qc/a, ax, qx/a, cd, dx \end{matrix} \middle| q \right]_\infty
$$
$$
\times \sum_{i,j\geq 0} \left[ \begin{matrix} \lambda/a, \lambda a/q \\ \lambda c, \lambda x \end{matrix} \middle| q \right]_{i+j} \left[ \begin{matrix} cd, dx \\ q, \lambda d \end{matrix} \middle| q \right]_i q^i \frac{(\lambda/d; q)_j}{(q; q)_j} \left( \frac{qcdx}{\lambda} \right)^j .
$$

By evaluating the last sum by (30) and then simplifying the result, we obtain the following expression in terms of well-poised series:

$$
\int_0^\pi \frac{h(\cos 2\theta; 1) h(\cos \theta; \lambda, bV/U)}{h(\cos \theta; a, q/a, b, c, d, x)} \prod_{k=1}^n \frac{h(\cos \theta; u_k)}{h(\cos \theta; v_k)} d\theta
$$

$$
= 2\pi \begin{bmatrix} \lambda c, \lambda d, \lambda x, acdx, qcdx/a, abV/U, qbV/aU \\ q, ab, ac, ad, ax, qb/a, qc/a, qd/a, qx/a, cd, cx, dx, \lambda cdx \end{bmatrix} q \Bigg]_\infty \tag{43}
$$

$$
\times \prod_{\iota=1}^n \begin{bmatrix} au_\iota, qu_\iota/a \\ av_\iota, qv_\iota/a \end{bmatrix} q \Bigg]_\infty {}_8W_7(\lambda cdx/q : \lambda/a, \lambda a/q, cd, cx, dx; q).
$$

When $U = V$, this formula is consistent with the case $x = q/a$ of (26).

*7.5. Boosting the Third Numerator Parameter $\mu$ by $\mathcal{G}_\mu(\mathcal{E})$*

Alternatively, by applying $\mathcal{G}_\mu(\mathcal{E})$ to (42), we obtain

$$
\int_0^\pi \frac{h(\cos 2\theta; 1) h(\cos \theta; \lambda, \mu, bV/U)}{h(\cos \theta; a, q/a, b, d, x)} \prod_{k=1}^n \frac{h(\cos \theta; u_k)}{h(\cos \theta; v_k)} d\theta
$$

$$
= 2\pi \begin{bmatrix} abV/U, qbV/aU, \lambda d, \mu x, \mu/x \\ q, ab, qb/a, ad, qd/a \end{bmatrix} q \Bigg]_\infty \prod_{\iota=1}^n \begin{bmatrix} au_\iota, qu_\iota/a \\ av_\iota, qv_\iota/a \end{bmatrix} q \Bigg]_\infty
$$

$$
\times \sum_{i,j \geq 0} \begin{bmatrix} \lambda/a, \lambda a/q \\ q, \lambda d \end{bmatrix} q \Bigg]_i \frac{q^i (\mu/x)^j}{(q;q)_j (\mu x; q)_j} \mathcal{E}^j \begin{bmatrix} q^i \lambda x \\ ax, qx/a, q^i dx \end{bmatrix} q \Bigg]_\infty,
$$

which gives rise to the following $q$-beta integral formula:

$$
\int_0^\pi \frac{h(\cos 2\theta; 1) h(\cos \theta; \lambda, \mu, bV/U)}{h(\cos \theta; a, q/a, b, d, x)} \prod_{k=1}^n \frac{h(\cos \theta; u_k)}{h(\cos \theta; v_k)} d\theta
$$

$$
= \begin{bmatrix} abV/U, qbV/aU, \lambda d, \lambda x \mu x, \mu/x \\ q, ab, ad, qb/a, qd/a, ax, dx, qx/a \end{bmatrix} q \Bigg]_\infty \prod_{\iota=1}^n \begin{bmatrix} au_\iota, qu_\iota/a \\ av_\iota, qv_\iota/a \end{bmatrix} q \Bigg]_\infty \tag{44}
$$

$$
\times 2\pi \sum_{i,j \geq 0} \frac{(dx;q)_{i+j}}{(\lambda x; q)_{i+j}} \begin{bmatrix} \lambda/a, \lambda a/q \\ q, \lambda d \end{bmatrix} q \Bigg]_i q^i \begin{bmatrix} ax, qx/a \\ q, \mu x \end{bmatrix} q \Bigg]_j \left(\frac{\mu}{x}\right)^j.
$$

Unfortunately, it seems impossible to reduce the above double series further to a single one.

Concluding Comments

The Askey–Wilson integral is fundamental in special functions and orthogonal polynomials. During the past two decades, many efforts have been made to generalize this important integral. In particular, two operators $\mathcal{P}_\lambda(\mathcal{D})$ and $\mathcal{Q}_\lambda(\delta)$ (based on the derivative operator $\mathcal{D}$) have been widely utilized by different authors to provide extensions. In the present paper, we reviewed the main results obtained by making use of these operators and commented on their strengths and weaknesses. By introducing a new operator $\mathcal{G}_\lambda(\mathcal{E})$, several remarkable integral formulae of Askey–Wilson type were also established.

According to the author's experience, one should take precautions, in applications of these operators, to conduct rigorous operations (instead of only formal manipulations) under the right conditions in order to avoid unexpected errors. In general, the sufficient conditions to make such operations legitimate (in exchanging orders between limit and summation/integration) are normally provided by Lebesgue's dominated convergence theorem (cf. [41] (§11.32)) requiring the sequence of functions involved to be bounded (for sums, and uniformly convergent for integrals).

Here, we take an example to illustrate the importance of observing the bounded condition in applications. Recall the operator $\mathcal{Q}_\lambda(\delta)$ defined in (6). Chen and Liu [21] proved the following useful formula:

$$\mathcal{Q}_\lambda(\delta)[ax, bx; q]_\infty = \frac{[ax, bx, a\lambda, b\lambda; q]_\infty}{(abx\lambda/q; q)_\infty}. \tag{45}$$

However, they failed to highlight the condition $|abx\lambda/q| < 1$, which is indispensable for the validity of the above formula. From exchanging the order of summations in their proof, the informed reader can retrieve the above condition. This operator $\mathcal{Q}_\lambda(\delta)$ was erroneously utilized by Zhang [42] in generalizing the following reciprocal formula due to Andrews [43]:

$$(b-d)\begin{bmatrix} q, & qb/d, & qd/b, & bcdx, & bdex, & bcde \\ bx, & dx, & bc, & be, & cd, & de \end{bmatrix} q \Big]_\infty$$
$$= b\sum_{n=0}^\infty \frac{[q/dx, bcde; q]_n}{[bc, be; q]_{n+1}}(bx)^n - d\sum_{n=0}^\infty \frac{[q/bx, bcde; q]_n}{[cd, de; q]_{n+1}}(dx)^n.$$

In order to apply (45), the above equality can be reformulated as

$$(b-d)\begin{bmatrix} q, & qb/d, & qd/b, & bcde \\ bc, & be, & cd, & de \end{bmatrix} q \Big]_\infty [bcdx, bdex; q]_\infty$$
$$= b\sum_{n=0}^\infty \frac{(bcde; q)_n}{[bc, be; q]_{n+1}} q^{\binom{n+1}{2}}\left(-\frac{b}{d}\right)^n [bx, q^{-n}dx; q]_\infty$$
$$- d\sum_{n=0}^\infty \frac{(bcde; q)_n}{[cd, de; q]_{n+1}} q^{\binom{n+1}{2}}\left(-\frac{d}{b}\right)^n [q^{-n}bx, dx; q]_\infty.$$

Ignoring the fact that $|q^{-n-1}\lambda bdx| < 1$ does not hold for all $n \in \mathbb{N}$, Zhang [42] formally proceeded with the application of (45) to both sides of the above equation and deduced the following false reciprocal identity:

$$(b-d)\begin{bmatrix} q, qb/d, qd/b, bcde, bcdx, bdex, bcd\lambda, bde\lambda, bdx\lambda/q \\ bc, be, cd, de, bx, dx, b\lambda, d\lambda, b^2cd^2ex\lambda/q \end{bmatrix} q \Big]_\infty$$
$$= b\sum_{n=0}^\infty \frac{[q/dx, q/d\lambda, bcde; q]_n\, q^n}{(q^2/bdx\lambda; q)_\infty [bc, be; q]_{n+1}} - d\sum_{n=0}^\infty \frac{[q/bx, q/b\lambda, bcde; q]_n\, q^n}{(q^2/bdx\lambda; q)_\infty [cd, de; q]_{n+1}}.$$

In fact, the correct formula reads as

$$b\sum_{n=0}^\infty \frac{[q/dx, q/d\lambda, bcde; q]_n\, q^n}{(q^2/bdx\lambda; q)_\infty [bc, be; q]_{n+1}} - d\sum_{n=0}^\infty \frac{[q/bx, q/b\lambda, bcde; q]_n\, q^n}{(q^2/bdx\lambda; q)_\infty [cd, de; q]_{n+1}}$$
$$= (b-d)\begin{bmatrix} q, qb/d, qd/b, bcde, bcdx, bdex, bcd\lambda, bde\lambda, bdx\lambda/q \\ bc, be, cd, de, bx, dx, b\lambda, d\lambda, b^2cd^2ex\lambda/q \end{bmatrix} q \Big]_\infty$$
$$+ \frac{bdx\lambda}{q}\begin{bmatrix} q, & q/bx, & q/dx, & q/b\lambda, & q/d\lambda, & bcde \\ bc, & be, & cd, & de, & q^2/bdx\lambda, & b^2cd^2ex\lambda/q \end{bmatrix} q \Big]_\infty$$
$$\times \left\{ b\begin{bmatrix} cd, de, b^2cdx\lambda, b^2dex\lambda \\ bx, b\lambda, q/bx, q/b\lambda \end{bmatrix} q \Big]_\infty {}_3\phi_2\begin{bmatrix} bx, b\lambda, b^2cd^2ex\lambda/q \\ b^2cdx\lambda, b^2dex\lambda \end{bmatrix} q; q \Big] \right.$$
$$\left. - d\begin{bmatrix} bc, be, bcd^2x\lambda, bd^2ex\lambda \\ dx, d\lambda, q/dx, q/d\lambda \end{bmatrix} q \Big]_\infty {}_3\phi_2\begin{bmatrix} dx, d\lambda, b^2cd^2ex\lambda/q \\ bcd^2x\lambda, bd^2ex\lambda \end{bmatrix} q; q \Big] \right\},$$

which was derived by Chu and Zhang [44] by employing the three-term relation [2] (Equation III-36) for nonterminating ${}_8\phi_7$-series.

**Funding:** This research received no external funding.

**Data Availability Statement:** Not applicable.

**Conflicts of Interest:** The author declares no conflict of interest.

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
