# Peer review of "The Askey–Wilson Integral and Extensions"

_mathematics, doi:10.3390/math11071759_

Round 1
Reviewer 1 Report
The paper under review deals with extensions of the Askey-Wilson integral. The results are interesting, mathematically correct, and the paper is well written, hence it merits publication in Mathematics. However, I have the following comments that will considerably improve the manuscript:
1. It is unclear what results are original from this work and what other results are revisions of known results. This should be explicitly specified.
2. The author basically evaluates different integrals by making use of different operators. No theorems are stated nor proved throughout the paper. This is why it is important to re-structure the manuscript. For instance, the second section is a revision of the q-derivative. No new results are provided in the second section. Therefore, this section should be renamed as Methodology or Background or Preliminaries or something similar.
3. In the third section, the author reviews and evaluates three q-beta integrals (Askey-Wilson integral and extensions) by means of the operator Pλ(D) (there is a misprint in the second line of the first paragraph of Section 3). So I assume that the original results begin with the third section.
4. Same for sections 4 and 5, where the operator Gλ(E) is used. Maybe, it would be more convenient to define and introduce in the second section all used operators to evaluate the integrals.
5. In the Conclusion section, the last sentence is somehow discouraging. Could the author provide a rigorous application of any of the above operators which leads to errors if the right assumptions are not assume? What could those assumptions be? I’m asking these questions because if strong assumptions are needed to apply the previous operators, then that might go against the interest of the non-experts on the field.
6. Finally, it would be interesting to show some applications of the evaluations of the integrals in other areas of Mathematics, such as for instance, Orthogonal Polynomials or Operator Theory (Functional Analysis).
Author Response
SEE UPLOAD PDF-FILE

Reviewer 2 Report
see the attached file.

Author Response
SEE UPLOAD PDF-FILE
